# Strengths and Weaknesses of Everyday Financial Knowledge and Judgment Skills of Adults with ADHD

**DOI:** 10.3390/ijerph20054656

**Published:** 2023-03-06

**Authors:** Janneke Koerts, Dorien F. Bangma, Christian Mette, Lara Tucha, Oliver Tucha

**Affiliations:** 1Department of Clinical and Developmental Neuropsychology, University of Groningen, 9712 TS Groningen, The Netherlands; 2Novicare, 3542 AB Utrecht, The Netherlands; 3Department of Social Work and Education, Protestant University of Applied Sciences Bochum, 44803 Bochum, Germany; 4LVR Hospital Essen, 45147 Essen, Germany; 5Faculty of Medicine, Department of Psychiatry and Psychotherapy, University of Duisburg-Essen, 45147 Essen, Germany; 6Department of Psychiatry and Psychotherapy, University Medical Center Rostock, 18147 Rostock, Germany; 7Department of Psychology, Maynooth University, National University of Ireland, W23 F2K8 Maynooth, Ireland

**Keywords:** ADHD, financial skills, financial knowledge, financial judgment, financial competence, financial capability, income

## Abstract

Adequate financial capability is crucial in everyday life. This capability might, however, not be given to adults with ADHD. The present study aims to determine strengths and weaknesses regarding everyday financial knowledge and judgment skills in adults with ADHD. In addition, the impact of income is explored. Forty-five adults with ADHD (M_age_ = 36.6, SD_age_ = 10.2) and 47 adults without ADHD (M_age_ = 38.5, SD_age_ = 13.0) were included and were assessed with the Financial Competence Assessment Inventory. Adults with ADHD showed decreased scores regarding awareness of the arrival of bills, knowledge of own income, having a reserve fund for unexpected expenses, the ability to state long-term financial goals, own preferences for estate management, understanding of assets, legal action for debt, access to financial advice/counseling, and the ability to compare medical insurance plans as compared with adults without ADHD (all *p* < 0.001). However, no effect of income was found. In conclusion, adults with ADHD have difficulties with many aspects of everyday financial knowledge and skills, which might result in a plethora of personal and legal consequences. It is, therefore, of the utmost importance that professionals who support adults with ADHD proactively ask about everyday financial functioning so that assessment, financial support, and coaching can be provided.

## 1. Introduction

Attention-deficit/hyperactivity disorder (ADHD) is a neurodevelopmental disorder that has a prevalence of approximately 3.4% in adulthood [1]. Numerous studies indicate that ADHD is associated with impairments in cognition, including attention, memory, speed of information processing, and executive functioning [2,3,4,5,6]. Furthermore, adults with ADHD often experience difficulties with affective functioning, including symptoms of depression, anxiety, and stress [7]. Together, these difficulties in cognitive and affective functioning can have a negative impact on functioning in the everyday lives of adults with ADHD, including educational, occupational, social, and financial functioning [8,9,10,11].

In our everyday lives, we make numerous financial decisions (including paying bills, budgeting, taking out insurance, etc.), and an adequate capability to make financial decisions is crucial for independent living, leisure-time activities, and social relations. Financial incapability can, therefore, have far-reaching personal and legal consequences and may lead to financial insecurity, debt, placement under financial guardianship, poverty, reduced social and societal participation, and diminished opportunities for leisure-time activities. Financial capability includes both financial competence and financial performance [12]. Financial competence can be assessed in a controlled setting and refers to the knowledge (i.e., ‘knowing what’ and ‘knowing how’) and judgment skills (i.e., the ability to identify and understand financial information, the ability to appreciate different options and reason about them, and the ability to express a choice [13]) needed to make financial decisions that serve one’s personal needs and goals. Financial performance refers to the actual degree of success in dealing with the financial demands, issues, or questions of everyday life, which is influenced by a range of contextual risk and protective factors (e.g., social support or substance use). As such, financial capability is an umbrella term that relies strongly on the integrity of various cognitive and affective functions [12,14,15].

Surprisingly, the financial capability of adults with ADHD is a topic that has received little attention so far. Indeed, our previous study [16] was the first to explore the financial capability of adults with ADHD in a standardized, objective manner. Up until then, studies mostly applied self-report measures and interviews. The scientific studies that have been performed have reported that adults with ADHD have lower income levels and savings–income ratios, are more often financially dependent, have more self-reported financial problems, have difficulties saving money, more often exceed credit card limits, and more often buy on impulse [17,18,19,20,21] as compared with adults without ADHD. In addition, adults with ADHD showed difficulties with financial judgment and particularly with understanding relevant financial information [22]. Importantly, the studies listed here, which were all performed on selected samples, are corroborated by a large-scale Swedish population study using objective financial data. This study demonstrated that adults who were ever diagnosed with ADHD were more likely to incur arrears, which appeared to be related to receiving new consumer credit less often, despite asking for more credit, than the general population. Particularly, adults with ADHD were four times more likely to accumulate arrears regarding unpaid educational support, road taxes, and alimony, misuse of bank accounts (e.g., overdrafts), and impounding of property as compared with the general population [23].

To deal with financial difficulties, general education programs aimed at improving financial knowledge and skills are often offered and are considered to be an antidote to dealing with financial issues and financial complexity by many policymakers [24]. A recent systematic review and meta-analysis, however, demonstrated that a one-size-fits-all approach to financial education does not result in improved financial behavior. The results of this meta-analysis showed that interventions in financial education only had weak effects on the financial behaviors that were studied, which, according to the authors, might be explained in two different ways. First, the effects of a financial education intervention might decay over time, which makes it difficult to apply knowledge acquired at one point in time to a current personal financial decision. Indeed, the meta-analysis demonstrated that decay over time was more pronounced for interventions with a longer duration. Second, the weak effects of interventions on financial behavior might be explained by the fact that financial education might not focus on the behavior or knowledge that is underlying one’s financial behavior [24]. The results of this meta-analysis thus indicate that to offer effective financial support and coaching, a more tailored approach is needed. To achieve this, it is crucial to determine the financial strengths and weaknesses that are associated with certain conditions and ultimately with an individual suffering from these conditions. The aim of the present study is, therefore, to determine strengths and weaknesses regarding the everyday financial knowledge and judgment skills of adults with ADHD. For this purpose, we will analyze the data from our previous study on financial decision-making in adults with ADHD [16] in more detail. Our previous study provided a helicopter view by examining many aspects of financial capability and related contextual factors in adults with ADHD. We could demonstrate that this group showed more difficulties with financial competence, impulsive buying, and decision-making with implications for the future than adults without ADHD. In the present study, we will particularly examine the construct of financial competence in more detail. The reason for focusing on financial competence is that it is a fundamental aspect of financial capability [12], as also mentioned above. In our previous study, financial competence was assessed with the Financial Competence Assessment Inventory (FCAI), and performance on this test was reported in terms of summary scores [25]. The FCAI is of much interest as it provides a large amount of information since it focuses on everyday financial knowledge and judgment skills and also partly examines the participant’s financial situation. The present study will thus examine the financial situation, financial knowledge, and judgment skills of adults with ADHD compared with those of adults without ADHD. The current research will advance knowledge on the financial capability of adults with ADHD by examining many aspects of everyday financial knowledge and skills in an objective and standardized manner, whereas previous research either used subjective information, examined only relatively few aspects of everyday financial capability, or used rather experimental tasks that might not have good ecological validity. In addition, as it has been suggested that income level has an influence on financial knowledge and skills [26,27], the present study will be the first to explore to what extent income level affects the financial situation, financial knowledge, and the judgment skills of adults with ADHD. It is hypothesized that adults with ADHD have a poorer financial situation than adults without ADHD and show difficulties with bill payment, deal more often with legal action for debt, and experience problems with understanding and judging financial information. Finally, it is hypothesized that lower income levels have a negative influence on the financial knowledge and skills of adults with ADHD.

## 2. Material and Methods

### 2.1. Study Design and Participants

The present study has a non-experimental, cross-sectional design. Adults with ADHD were recruited via the Department of Psychiatry and Psychotherapy at the University of Duisburg-Essen, Germany. All adults with ADHD were diagnosed according to the criteria of the Diagnostic and Statistical Manual of Mental Disorders, 5th edition [28]. Adults without ADHD were contacted via the researchers, social media, and word-of-mouth.

Data were collected between 2014 and 2016. Participants were assessed individually by research assistants who were all extensively trained. The use of a detailed assessment and scoring manual guaranteed consistent assessment and scoring.

All participants were requested to complete the Wender Utah Rating Scale—Childhood (WURS-K [29,30]), which includes the retrospective assessment of childhood symptoms of ADHD, and the ADHD self-report scale (ADHD-SR [30]), which assesses current symptoms of ADHD. Adults with ADHD were excluded when they scored below the cut-off on both questionnaires (i.e., WURSK < 30 and ADHD-SR < 18 [30]), whereas adults without ADHD were excluded when they scored on or above the cut-off on both questionnaires (i.e., WURS-K ≥ 30 and ADHD-SR ≥ 18 [30]). Please see below for a more detailed description of these questionnaires. Furthermore, participants were excluded when diagnosed with a severe neurological or psychiatric condition other than ADHD. Within this context, common comorbidities of ADHD were not considered reasons for exclusion. Within the context of the present study, participants were finally excluded when their scores on the FCAI were not available at the item level. The latter resulted in the exclusion of four additional adults without ADHD as compared with our previous study [16]. In the present study, data from 45 adults with ADHD (M_age_ = 36.6, SD_age_ = 10.2, range 19–61 years) and 47 adults without ADHD (M_age_ = 38.5, SD_age_ = 13.0, range 20–64 years) were used for analyses.

### 2.2. Material

The following demographic information was collected on all participants: age (in years), years of education, sex (male/female), and work status (full-time, part-time, unemployed, or other).

The FCAI [25] is a standardized, comprehensive instrument that assesses a broad range of financial skills in everyday life. The FCAI items have high internal consistency (r = 0.96), excellent inter-rater reliability (89%), and good content, convergent, and concurrent validity [25]. All 38 items of the FCAI are described in Table 1. Unless mentioned otherwise, all items include open-ended questions that are scored on a five-point scale ranging from 0 = no awareness, 1 = rudimentary awareness, 2 = partial understanding, 3 = adequate understanding, to 4 = complete understanding.

Annual gross income of participants was rated on a six-point scale: (1) <EUR 15,000, (2) EUR 15,000–EUR 25,000, (3) EUR 25,000–EUR 35,000, (4) EUR 35,000–EUR 45,000, (5) EUR 45,000–EUR 55,000, and (6) >EUR 55,000.

Symptoms of ADHD were assessed with the WURS-K and ADHD-SR. The WURS-K is a self-report questionnaire that consists of 25 items focused on childhood symptoms of ADHD, which are rated on a 5-point scale ranging from 0 (i.e., not at all or very slightly), 1 (i.e., mildly), 2 (i.e., moderately), 3 (i.e., quite a bit), to 4 (i.e., very much). The WURS-K was found to have high internal consistency (r = 0.91), inter-rater reliability (r = 0.90), and good convergent validity (r = 0.94) [29,30]. Current symptoms of ADHD were assessed with the ADHD-SR, which contains 18 items that are rated on a 4-point scale that ranges from 0 (i.e., does not apply), 1 (i.e., mildly present), 2 (i.e., moderately present), to 3 (i.e., very present). The ADHD-SR has good validity, inter-rater reliability (between r = 0.78 and r = 0.89) and internal consistency (between r = 0.72 and r = 0.90) [30].

### 2.3. Statistical Analyses

The Statistical Package for the Social Sciences, version 28.0. IBM, New York, NY, USA, was used for analyses. Not all continuous variables (i.e., age, years of education, WURS-K, and ADHD-SR) were normally distributed. Therefore, Mann–Whitney U tests were used for group comparison. Chi-square tests were used to compare adults with and without ADHD on categorical variables.

To determine whether adults with and without ADHD showed different scores on the items of the FCAI, which are measured on an ordinal level, Mann–Whitney U tests were used for comparison. Subsequently, all participants were allocated to one of two income groups: (1) people with a relatively low income (i.e., earning an annual gross income of less than EUR 35.000) or (2) people with a relatively high income (i.e., earning an annual gross income of more than EUR 35.000). To explore to what extent income affects scores on the items of the FCAI, Mann–Whitney U tests were used to compare people with a relatively low and a relatively high income. The latter analysis was conducted in groups of adults with and without ADHD separately.

Effect sizes (Cohen’s d) were calculated for all comparisons of scores on FCAI items and interpreted as small (d = 0.2), medium (d = 0.5), and large (d = 0.8) [31]. To correct for multiple comparisons, a Bonferroni correction was applied. A *p*-value of 0.05/38 = 0.001 was, therefore, considered significant. Finally, participants with missing data were excluded pairwise, i.e., a participant was omitted from the analysis of item 1 and included in the analysis of item 2 if they had a missing value for item 1 but not for item 2.

## 3. Results

The clinical and descriptive characteristics of both groups are presented in Table 2. Both groups had a similar age and completed a similar number of years of education. Furthermore, no differences were found between groups regarding the number of males and females. As expected, adults with ADHD showed significantly higher scores on the WURS-K and ADHD-SR than adults without ADHD (Table 2).

Adults with ADHD showed poorer scores on several items of the FCAI as compared with adults without ADHD (Table 3 and Figure 1). Significant differences were specifically found for awareness of the arrival of bills (item 2), knowledge of own income (item 9), a reserve fund for unexpected expenses (item 12), the ability to state long-term financial goals (item 15), own preferences for estate management (item 16), understanding of assets (item 20), legal action for debt (item 23), access to financial advice/counseling (item 26), and the ability to compare medical insurance plans (item 29). These differences are all consistent with the medium-to-large effect sizes that were found (Table 3). For all other items of the FCAI, no significant differences were found between adults with and without ADHD.

Furthermore, adults with ADHD had a significantly lower income than adults without ADHD (Table 2). While adults without ADHD had a median annual gross income between EUR 35,000 and EUR 45,000, adults with ADHD reported having a median annual gross income between EUR 15,000 and EUR 25,000. To explore to what extent income level was of influence on the scores on items of the FCAI, participants were allocated to a relatively low-income group (i.e., annual gross income less than EUR 35,000; adults with ADHD *n* = thirty-one; adults without ADHD *n* = twenty-one); or a relatively high-income group (i.e., annual gross income more than EUR 35,000; adults with ADHD *n* = thirteen; adults without ADHD *n* = twenty-four; income data were unknown for three participants). Within the group of adults with ADHD, no significant differences were found between the relatively low and relatively high-income groups (Table 4). Similar results were found within the group of adults without ADHD, i.e., adults with a relatively low or a relatively high income showed comparable scores on the FCAI (Table 4).

## 4. Discussion

The present study aimed to determine strengths and weaknesses regarding everyday financial knowledge and judgment skills in adults with ADHD. The results of our study show that adults with ADHD have difficulties with several aspects of everyday finances as compared with adults without ADHD, findings that are in line with previous studies that focused on financial difficulties in adults with ADHD [17,18,19,20,21,23]. Our results specifically indicate that adults with ADHD have a reduced knowledge of their own income and a reduced ability to state long-term financial goals as compared with adults without ADHD. Furthermore, adults with ADHD are less aware of the arrival of bills and/or have more difficulties recognizing them, set aside less money for unexpected expenses, have a decreased understanding of their assets, and have more difficulty comparing medical insurance plans than adults without ADHD. It is worrisome that adults with ADHD show a reduced performance on these specific aspects of everyday finances as this might directly result in financial problems, for example, when bills are paid too late or not at all, when insufficient or no money is available because of not having an overview of how much money is coming in each pay period, when there are no clear financial goals, or when there is no reserve fund for unexpected expenses. Furthermore, these difficulties with handling everyday finances can also have far-reaching consequences that are not financial in nature. For example, difficulties with comparing medical insurance plans may result in selecting a suboptimal insurance plan that might not only lead to increased medical costs but also to not receiving required medical care or receiving medical care too late when no money is available to pay the increased medical costs. Collectively, these findings are especially alarming considering the results of the large Swedish population study by Beauchaine and colleagues, using actual financial data and demonstrating that adults who were ever diagnosed with ADHD are more likely to incur arrears than adults without ADHD and that financial distress is associated with an increased suicide risk [23]. The data of the present study also show that adults with ADHD regularly end up in difficult financial situations, as adults with ADHD more often report that legal action was taken against them in the last five years for debt as compared with adults without ADHD. Indeed, according to the raw data, 27% (*n* = 12) of adults with ADHD had to deal with legal action for debt in the last five years, while this was only the case in 2% (*n* = 1) of adults without ADHD.

Currently, it is unclear what the underlying cause of these reduced financial performances in adults with ADHD is. It is, of course, likely that the cognitive and affective symptoms that accompany ADHD play a role. Indeed, according to the dual-pathway model of decision-making, both motivational/affective and deliberative/analytic processing of information are involved in the decision-making process, including financial decision-making [32,33,34]. Since the deliberative/analytical processing of information relies strongly on executive functioning, attention, and memory, and because many studies have demonstrated that adults with ADHD have impairments in these cognitive functions [2,3,5,6] and in affective functioning [7,35,36], these symptoms of ADHD might take their toll on financial competence and financial performance in everyday life. However, since ADHD is a neurodevelopmental condition, it might also be that adolescents and young adults with this condition have difficulties acquiring sufficient financial knowledge and the judgment skills needed for everyday life. In addition, it might be argued that the results found in the present study are due to the finding that adults with ADHD have a significantly lower income than adults without ADHD. Because of this lower income, possibly in combination with arrears, adults with ADHD might have fewer opportunities to create and maintain a reserve fund for unexpected expenses or have few assets and are, therefore, unable to state what their major assets are. Previous studies also reported that income might have an influence on financial knowledge and skills [26,27]. The results of the present study, however, indicate that there are no differences between those with a relatively low (i.e., annual gross income of less than EUR 35,000) and those with a relatively high income (i.e., annual gross income of more than EUR 35,000) regarding the scores on the items of the FCAI, neither within the group of adults with ADHD nor within the group of adults without ADHD. Within the group of adults with ADHD, there were, however, many items such as help with estate/money management, awareness of the arrival of bills, understanding banking protocols, the ability to budget, and the ability to state long-term financial goals that were associated with medium or medium-to-large effect sizes, indicating significant differences might be found between relatively low and high income groups when larger groups would have been examined. Furthermore, the cut-off of EUR 35,000 is, of course, arbitrary. Different results might have been found when comparing groups with an annual gross income of less than EUR 25,000 and more than EUR 55,000. This comparison was, however, not feasible considering the size of our samples.

The results of the present study further indicate that adults with ADHD more often state that they would like for someone else to make their financial decisions or that they need assistance when making financial decisions (i.e., their own preferences for estate management). When exploring the exact answers of adults with and without ADHD regarding this item, it becomes clear that no fewer than 33% (*n* = 15) of adults with ADHD would like for someone else to make their financial decisions or that they require assistance when making financial decisions, compared with 2% (*n* = 1) of adults without ADHD. Furthermore, of the adults with ADHD who require financial assistance or would like for someone else to take over their financial decisions, 47% (*n* = 7) have had to deal with legal action for debt in the last five years. These findings are disturbing, as the results of the present study also show that adults with ADHD more often state that they have less access to these services than adults without ADHD. The latter finding might indicate that adults with ADHD are less aware or have less knowledge about financial constructs, services, protocols, etc., and where to find them—an assumption that would be in line with the finding that adults with ADHD have decreased knowledge of what assets are and have difficulties naming their major assets compared with adults without ADHD. This assumption is, however, not supported by the data since adults with and without ADHD scored similarly on items such as knowledge of programs/services offering financial support, understanding banking protocols, and understanding credit. The discrepancy between the financial support adults with ADHD receive or would like to have and their actual access to financial services thus requires further investigation to determine its underlying cause.

When exploring the results of the present study, it is important to keep in mind that there were many items on the FCAI for which no significant differences were found between adults with and without ADHD. This indicates that there are also many aspects of financial knowledge and judgement skills, including basic arithmetic skills, the ability to differentiate between medical insurance plans and to complete the associated form, the ability to prevent being disconnected from services due to non-payment or the repossession of goods to settle debts, and understanding items on a bank statement, on which adults with and without ADHD show similar scores; comparisons that were accompanied by negligible or small effect sizes. It is, however, not clear whether all the items for which no significant differences were found between groups can be considered financial strengths for adults with ADHD. The reason is that several of the items for which no significant group differences were found, including the ability to read a household bill, write a bank transfer, knowledge of fixed monthly costs, the ability to budget, and financial assistance-seeking skills, are accompanied by medium-to-large effect sizes. As all these aspects of everyday finances can result in financial difficulties, future studies with larger sample sizes must explore whether adults with ADHD show reduced scores or not on these items compared with adults without ADHD to shed more light on which aspects of everyday financial knowledge and judgment skills can be considered strengths and which aspects are weaknesses. This information can support providing psychoeducation for those skills that are suboptimal, or it can be used to adapt existing training and coaching programs so that they are tailored to the needs of a specific population or a specific individual.

The main strength of the present study was that it was one of the few that examined many aspects of everyday financial knowledge and skills in an objective and standardized manner in adults with ADHD. Furthermore, the task that was used (i.e., the FCAI) has good content and concurrent validity [25], and the present study is the first to explore the influence of income level on the financial situation, financial knowledge, and judgment skills of adults with ADHD. Despite these strengths, there are still many questions that remain unanswered. Future research is needed to determine the underlying cause and/or supporting factors of financial difficulties in adults with ADHD, including the impact of cognitive and affective functioning as well as the influence of income. Furthermore, it must be determined whether tailored support or advice can be useful and improve the financial situation, skills, and knowledge of adults with ADHD. Finally, it is crucial to determine whether adults with ADHD need preventive measures to lessen financial difficulties and financial stress.

Within the context of the present study, several limitations/weaknesses must also be considered. First, even though the FCAI assesses a large number of aspects of everyday financial skills, there are still many aspects of everyday financial capability that were not considered in the present study. For example, the ability to compare financial products (e.g., mortgages) other than insurance, knowledge about taxes, and the ability to detect financial deception or fraud were not examined. Since aspects such as these can also negatively influence everyday financial functioning, they require further exploration in future studies. A second limitation is that the researchers of the present study were aware of the diagnoses of the participants, which might have resulted in an observer bias. Even though a standardized protocol was used for assessment and scoring, which should have minimized this effect, it cannot be excluded. Third, a larger sample size is needed to reproduce our findings and provide more reliable results. Indeed, a posthoc power analysis (using G*Power [37,38]) revealed that sufficient power (i.e., ≥0.80) was achieved in the main analysis only for effect sizes of 0.87 or larger, while the comparisons of the relatively low- and high-income groups did not yield sufficient power. Finally, it is important to keep in mind that the present study did not make use of actual financial data. This might have resulted, in both groups, in an under- as well as an overestimation of financial knowledge and judgment skills. Specifically, the use of self-reported data for income level, combined with the finding that adults with ADHD have a decreased knowledge of their own income compared with adults without ADHD, might have influenced the comparison of the relatively low and relatively high-income groups.

## 5. Conclusions

The results of the present study show that adults with ADHD have difficulties with many aspects of everyday financial knowledge and skills. These financial difficulties might result in a plethora of personal and legal consequences, such as financial insecurity, debts, placement under financial guardianship, poverty, reduced social and societal participation, and diminished opportunities for leisure-time activities. These findings are alarming, as no fewer than one-third of adults with ADHD state that they would like assistance or support with making financial decisions, 27% have had to deal with legal action for debt in the last five years, and adults with ADHD indicate that they have less access to financial assistance than adults without ADHD. It is, therefore, of the utmost importance that professionals who support adults with ADHD not only focus on the symptoms and treatment but also proactively ask about everyday functioning, including everyday financial functioning. Only in this way can an assessment of financial knowledge and skills be conducted with tests such as the FCAI, and tailored financial support and coaching be provided.

## Figures and Tables

**Figure 1 ijerph-20-04656-f001:**
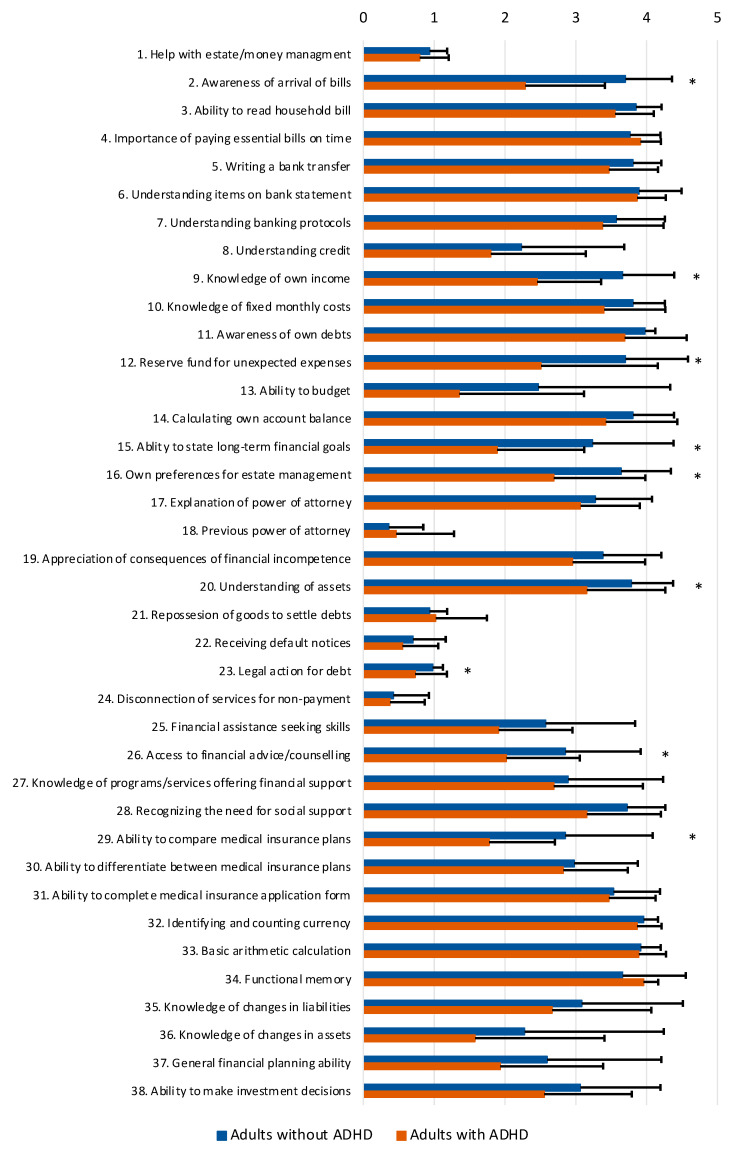
Comparison of performance on FCAI items (M and SD) of adults with (*n* = 45) and without ADHD (*n* = 47). Note: * *p* < 0.001.

**Table 1 ijerph-20-04656-t001:** Description of the items of the FCAI [25].

Items of the FCAI	Description of Items
1. Help with estate/money management	This item consists of a multiple-choice question about help with estate/money management that is scored on a two-point scale: 0 = receives help, 1 = receives no help.
2. Awareness of arrival of bills	This item includes questions related to the arrival of bills and how to recognize them.
3. Ability to read household bills	During this item, participants are presented with a typical household bill, about which questions are asked (e.g., when is the bill due and how much should be paid).
4. Importance of paying essential bills on time	This item includes the presentation of an electricity bill. Questions are asked about whether it is important to pay the bill on time and what might happen if the bill is not paid on time.
5. Writing a bank transfer	During the item, participants have to write a bank transfer for the electricity bill of item 4.
6. Understanding items on bank statement	Participants are presented with two different bank statements, depending on which questions are asked. For example, are there sufficient funds to pay the electricity bill, and which column shows the money that is coming in?
7. Understanding banking protocols	Questions are asked related to overdraft, i.e., what would happen if there were insufficient funds to pay for the electricity bill and there was an overdraft facility on the bank account?
8. Understanding credit	Questions are asked about whether participants understand that using credit is related to additional costs.
9. Knowledge of own income	This item includes questions about whether participants are aware of their sources of income and how much income they receive.
10. Knowledge of fixed monthly costs	Questions are asked to determine whether participants are aware of where their money goes each month.
11. Awareness of own debts	Participants are asked whether they have debts that are paid by someone else (e.g., when receiving social welfare).
12. Reserve funds for unexpected expenses	This item includes questions about whether participants are keeping money on the side for unexpected expenses.
13. Ability to budget	Participants are asked questions related to whether they make sure that money is available for essentials (e.g., food, bills, clothes) when they receive their income.
14. Calculating own account balance	Participants are provided with paper and a pencil and are told that they have a certain amount on their bank account. Subsequently, they have to calculate what their account balance is after paying two different things.
15. Ability to state long-term financial goals	Participants are asked whether they would like to do something special with their money or if there is something they would like to achieve financially.
16. Own preferences for estate management	During this item, questions are asked about whether participants would like to make their own financial decisions or whether they would like someone else to decide.
17. Explanation of power of attorney	Participants are asked to explain what a power of attorney is.
18. Previous power of attorney	This item consists of a multiple-choice question about whether participants ever gave someone financial power of attorney, which is scored on a two-point scale: 0 = yes, 1 = no.
19. Appreciation of consequences of financial incompetence	Questions related to this item focus on what it might mean if someone is considered to be financially incompetent.
20. Understanding of assets	Participants are asked to explain what assets are and to name their major assets.
21. Repossession of goods to settle debts	This item consists of a multiple-choice question about whether participant’s goods were ever repossessed to settle debts and whether this happened in the last five years. This item is scored on a two-point scale: 0 = yes, within the last five years, 1 = no, or not within the last five years.
22. Receiving default notices	During this multiple-choice item, participants are asked if they received any default notices in the last five years. This item is also scored on a two-point scale: 0 = yes, within the last five years, 1 = no, or not within the last five years.
23. Legal action for debt	This item consists of a multiple-choice question about whether any legal action was taken against the participant in the last five years because of debt, which is scored as 0 = yes, within the last five years, or 1 = no, or not within the last five years.
24. Disconnection of services for non-payment	Participants are asked if they were ever disconnected from services such as gas and electricity due to non-payment and whether this happened in the last five years. This multiple-choice item is also scored on a two-point scale: 0 = yes, within the last five years, 1 = no, or not within the last five years.
25. Financial assistance-seeking skills	Questions are asked about where to look for financial assistance in case of problems managing money.
26. Access to financial advice/counseling	During this item, participants are asked where they would go for financial advice or counseling.
27. Knowledge of programs/services offering financial support	Participants are asked to tell more about programs or services that are available for people who have financial difficulties.
28. Recognizing the need for social support	This item consists of a multiple-choice question during which participants are asked to rate, on a five-point scale (0 = not important, 4 = very important), how important they think social support is.
29. Ability to compare medical insurance plans	Participants are presented with three medical insurance plans and asked which plan would best cover their dental costs.
30. Ability to differentiate between medical insurance plans	Participants are again presented with the three medical insurance plans. Subsequently, they are asked whether one of the health care plans covers a specific type of care.
31. Ability to complete medical insurance application form	Participants receive a medical insurance application form and are requested to complete it.
32. Identifying and counting currency	Participants are presented with several notes and coins and are asked to state the value of all notes and coins and to determine the total.
33. Basic arithmetic calculation	During this item, participants receive a menu from a restaurant and are told that they have a certain amount of money. Subsequently, questions are asked about what they can order from this menu.
34. Functional memory	Between items 14 and 15, participants are asked to state four numbers that they can remember (representing a PIN) and are told they need to recall them later. During the current item, they are again asked to state these four numbers.
35. Knowledge of changes in liabilities	Participants are asked whether there have been any changes in their financial liabilities. For example, when did the amount that is paid for rent or mortgage last change?
36. Knowledge of changes in assets	Participants are asked what would happen to the value of a house during a period of high inflation.
37. General financial planning ability	During this item, participants are asked whether they normally put money aside for expected bills.
38. Ability to make investment decisions	Participants are asked what they would do with a large lump sum of money that would need to support them.

**Table 2 ijerph-20-04656-t002:** Descriptive and clinical characteristics of adults with (*n* = 45) and without ADHD (*n* = 47).

	Adults with ADHD	Adults without ADHD	Statistic	*p*
	M (SD)	M (SD)		
Continuous variables				
**Age in years**	36.6 (10.2)	38.5 (13.0)	z = −0.49	0.628
**Education in years**	16.6 (3.3)	17.0 (4.0)	z = −0.25	0.805
**WURS-K ^#^**	42.9 (13.1)	13.2 (8.2)	z = 7.47	<0.001 *
**ADHD-SR ^#^**	35.6 (8.6)	11.1 (7.7)	z = 7.70	<0.001 *
Categorical variables				
**Sex: % Female ^#^**	42.2	47.8	X^2^ = 0.29	0.591
**Annual gross income *Median***	EUR 15.000–EUR 25.000	EUR 35.000–EUR 45.000	z = −3.27	0.001 *
% <EUR 15,000	40.9	13.3		
% EUR 15,000–EUR 25,000	11.4	13.3		
% EUR 25,000–EUR 35,000	18.2	20.0		
% EUR 35,000–EUR 45,000	15.9	8.9		
% EUR 45,000–EUR 55,000	4.5	13.3		
% >EUR 55,000	9.1	31.1		
**Work-status**			X^2^ = 1.54	0.673
% Full-time	51.1	61.7		
% Part-time	24.4	14.9		
% Unemployed	13.3	12.8		
% Other	11.1	10.6		
**Presentation of ADHD**				
% Combined	57.8	-		
% Inattentive	22.2	-		
% Hyperactive/Impulsive	0	-		
% Not specified	20.0	-		
**Comorbidities**				
% Adjustment disorder	15.6	-		
% Depressive disorder	13.3	-		
% Personality disorder	11.1	-		
% Substance dependency	6.7	-		

Note: WURS-K: Wender Utah Rating Scale—Childhood; ADHD-SR: ADHD self-report scale; ^#^ Missing data: Sex—without ADHD *n* = 1; WURS-K—with ADHD *n* = 4, without ADHD *n* = 1; ADHD-SR—with ADHD *n* = 3; * *p* < 0.001; groups were compared with Mann–Whitney U-tests or chi-square (X^2^) tests.

**Table 3 ijerph-20-04656-t003:** Comparison of performance on FCAI items of adults with (*n* = 45) and without ADHD (*n* = 47).

	z	*p*	d
1. Help with estate/money management	−2.10	0.036	0.41
2. Awareness of arrival of bills	−6.25	<0.001 *	1.54
3. Ability to read household bill	−2.92	0.003	0.64
4. Importance of paying essential bills on time	1.87	0.061	0.40
5. Writing a bank transfer	−2.59	0.010	0.60
6. Understanding items on bank statement	−1.20	0.231	0.05
7. Understanding banking protocols	−1.10	0.270	0.25
8. Understanding credit	−1.51	0.130	0.31
9. Knowledge of own income ^#^	−5.80	<0.001 *	1.47
10. Knowledge of fixed monthly costs	−2.76	0.006	0.59
11. Awareness of own debts	−2.29	0.022	0.46
12. Reserve fund for unexpected expenses	−4.31	<0.001 *	0.90
13. Ability to budget	−2.81	0.005	0.61
14. Calculating own account balance	−2.07	0.038	0.47
15. Ability to state long-term financial goals	−4.91	<0.001 *	1.13
16. Own preferences for estate management	−4.26	<0.001 *	0.91
17. Explanation of power of attorney	−1.23	0.219	0.26
18. Previous power of attorney	0.10	0.922	0.16
19. Appreciation of consequences of financial incompetence	−2.24	0.025	0.46
20. Understanding of assets	−3.38	<0.001 *	0.71
21. Repossession of goods to settle debts	−0.09	0.931	0.16
22. Receiving default notices	−1.45	0.148	0.30
23. Legal action for debt	−3.36	<0.001 *	0.74
24. Disconnection of services for non-payment	−0.46	0.642	0.10
25. Financial assistance seeking skills	−2.43	0.015	0.57
26. Access to financial advice/counseling	−3.49	<0.001 *	0.79
27. Knowledge of programs/services offering financial support	−1.24	0.217	0.16
28. Recognizing the need for social support	−3.00	0.003	0.68
29. Ability to compare medical insurance plans	−4.29	<0.001 *	0.98
30. Ability to differentiate between medical insurance plans	−0.85	0.393	0.17
31. Ability to complete medical insurance application form	−0.55	0.585	0.10
32. Identifying and counting currency	−1.54	0.124	0.32
33. Basic arithmetic calculation	−0.10	0.924	0.08
34. Functional memory	1.75	0.080	0.46
35. Knowledge of changes in liabilities	−1.82	0.068	0.29
36. Knowledge of changes in assets	−1.82	0.069	0.37
37. General financial planning ability	−2.54	0.011	0.43
38. Ability to make investment decisions	−2.11	0.035	0.43

Note: ^#^ Missing data: with ADHD *n* = 1; * *p* < 0.001. Mann–Whitney U tests were used for analysis.

**Table 4 ijerph-20-04656-t004:** Comparison of performance on FCAI items of adults with ADHD with a relatively low (*n* = 31) and a relatively high income (*n* = 13) and adults without ADHD with a relatively low (*n* = 21) and a relatively high income (*n* = 24).

	Adults with ADHD			Adults without ADHD		
	z	*p*	d	z	*p*	d
1. Help with estate/money management	2.15	0.031	0.74	−0.47	0.636	0.14
2. Awareness of arrival of bills	1.69	0.091	0.67	−0.52	0.605	0.21
3. Ability to read household bill	0.29	0.776	0.12	0.60	0.550	0.18
4. Importance of paying essential bills on time	1.34	0.179	0.54	2.66	0.008	0.84
5. Writing a bank transfer	−0.53	0.597	0.18	−0.15	0.883	0.04
6. Understanding items on bank statement	−0.93	0.353	0.28	1.53	0.126	0.38
7. Understanding banking protocols	1.89	0.059	0.73	1.87	0.062	0.54
8. Understanding credit	0.55	0.580	0.17	1.15	0.249	0.32
9. Knowledge of own income ^#^	0.29	0.254	0.45	1.30	0.192	0.36
10. Knowledge of fixed monthly costs	0.40	0.693	0.22	3.04	0.002	0.91
11. Awareness of own debts	1.01	0.312	0.46	1.07	0.285	0.31
12. Reserve fund for unexpected expenses	1.40	0.160	0.47	0.76	0.448	0.44
13. Ability to budget	−1.52	0.128	0.45	−0.51	0.607	0.19
14. Calculating own account balance	−1.25	0.210	0.43	0.58	0.559	0.12
15. Ability to state long-term financial goals	2.14	0.032	0.76	0.59	0.557	0.16
16. Own preferences for estate management	0.43	0.668	0.21	0.06	0.954	0.19
17. Explanation of power of attorney	0.56	0.572	0.19	1.08	0.282	0.35
18. Previous power of attorney	−1.07	0.285	0.44	−1.56	0.118	0.47
19. Appreciation of consequences of financial incompetence	0.78	0.437	0.36	0.42	0.673	0.10
20. Understanding of assets	−0.01	0.989	0.06	0.69	0.491	0.27
21. Repossession of goods to settle debts	−0.06	0.952	0.22	−1.66	0.097	0.52
22. Receiving default notices	0.41	0.686	0.13	0.94	0.349	0.28
23. Legal action for debt	−0.33	0.739	0.11	−0.94	0.350	0.29
24. Disconnection of services for non-payment	−0.49	0.621	0.16	−0.99	0.322	0.30
25. Financial assistance seeking skills	0.22	0.829	0.06	0.56	0.578	0.15
26. Access to financial advice/counseling	1.53	0.127	0.47	0.08	0.934	0.02
27. Knowledge of programs/services offering financial support	1.21	0.226	0.09	−0.30	0.764	0.15
28. Recognizing the need for social support	0.28	0.782	0.02	−0.24	0.808	0.17
29. Ability to compare medical insurance plans	1.90	0.058	0.63	0.48	0.631	0.16
30. Ability to differentiate between medical insurance plans	−0.34	0.735	0.11	1.93	0.054	0.58
31. Ability to complete medical insurance application form	0.41	0.683	0.13	0.73	0.464	0.24
32. Identifying and counting currency	0.74	0.462	0.26	1.53	0.126	0.45
33. Basic arithmetic calculation	0.23	0.816	0.14	0.71	0.477	0.21
34. Functional memory	0.93	0.354	0.37	0.08	0.939	0.12
35. Knowledge of changes in liabilities	1.85	0.065	0.65	0.93	0.352	0.37
36. Knowledge of changes in assets	1.88	0.060	0.63	1.87	0.061	0.56
37. General financial planning ability	−0.90	0.369	0.28	−0.13	0.895	0.07
38. Ability to make investment decisions	2.63	0.008	0.93	1.79	0.073	0.51

Note: ^#^ Missing data: with ADHD *n* = 1; *p*-values of <0.001 were considered significant; Mann–Whitney U tests were used for analysis.

## Data Availability

The data that support the findings of this study are available from the corresponding author upon reasonable request.

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
