# Peer review of "Strengths and Weaknesses of Everyday Financial Knowledge and Judgment Skills of Adults with ADHD"

_ijerph, 2023, doi:10.3390/ijerph20054656_

Round 1
Reviewer 1 Report
First, I would like to thank you for the opportunity to review this study, I found it interesting to read about it and review the literature.
The study aimed to evaluate the financial capability of adults with ADHD compared to those without. 45 adults with ADHD and 47 without were assessed using the Financial Competence Assessment Inventory. Results showed that adults with ADHD have decreased scores in several areas of everyday financial knowledge and skills, but no effect of income was found. The findings suggest that adults with ADHD have difficulties in managing their finances, which can lead to personal and legal consequences. It is important for professionals to provide financial support and coaching to adults with ADHD.
In general, the study presents some weaknesses that have already been pointed out by the authors themselves in the discussion, but I would like to share with the authors some questions/reflections before considering the publication.
-
Consider rearranging the presentation of the introduction to make the relationship between ADHD and financial capacity clearer.
-
Provide more details about the meta-analysis mentioned in the introduction, to reinforce the importance of a more personalized approach.
-
Review grammar and language clarity.
-
To improve the study, a larger sample size could be used to provide more robust results, and a more comprehensive investigation into the influence of income on financial knowledge and skills could be conducted. Additionally, a comparison between different income groups could be performed to provide further insight into this relationship.
-
Further research is needed to determine the underlying cause of financial problems in adults with ADHD.
-
It is important to consider that the cognitive and affective symptoms of ADHD can have an impact on financial decision making.
-
It is necessary to investigate whether financial education could improve financial skills and competence in adults with ADHD.
-
It is important to take into account that income may be an influential factor in financial competence, and wider income group comparisons should be considered in future research.
-
It is necessary to explore whether financial advice or assistance could be useful for adults with ADHD who wish to improve their financial skills.
-
It is important to consider the need for preventive measures to help prevent financial problems and financial stress in adults with ADHD.
I hope these reflections help you improve the discussion of the article and future lines of research.
Regards,
Reviewer 2 Report
Reviever Comment
The subject of the research was not very original in general. Manuscript was in many ways very far from an article writing format. The introduction part was mixed and a flow could not be achieved. At the end of the introduction, the purpose was not clearly stated and the hypothesis remained incomplete. Literature studies are rarely mentioned, and most importantly, why the study is important to the literature and how it differs from other studies and its importance is not clearly stated. The material method section was the most authoritative part of the manuscript and the design of the study was not written. A lot of information about the research was meaningless. The data to be given in the results are given in the participants section. No explanation has been given regarding the inventory used. The discussion section contained too many personal comments. A discussion is made with the literature and comments are made by establishing a relationship with the results of this literature. The strengths, weaknesses and limitations of the study are not given. In general, there are many deficiencies and areas that need to be corrected. Required regulations are listed below.
1. Page 1, line 18; Add your statistical significance values to the conclusion section of the summary.
2. Page 1, line 32; Some of the keywords are unnecessary. Choose keywords target-oriented. Also, put words of high importance: ADHD at the beginning. Delete the Everyday function.
3. Page 2, line 52; You must cite the source for any information you provide in the Manuscript. At the end of the paragraph, add the source of the article from which you received the information.
4. Page 2, line 55-59; Recent studies and clinical observations indicate that attention deficit/hyperactivity disorder (ADHD) is one of 56 the conditions that has a negative impact on the capability to make financial decisions. If you are using a plural expression like this, you should add the source of more than one work at the end of the sentence. Include these research resources.
5. Page 3, line 98-100; Include the source of the inventory you used at the end of the sentence.
6. Page 3, line 98-110; You do not need to mention the inventory you use so much in the introduction. Information about this inventory should be explained in detail in the method section. In addition, the aim of your research cannot be how the scores will be affected, you are evaluating the financial situation, not the scores. Points are the means, not the goal. This and other parts of the introduction should be arranged in detail from this perspective. A hypothesis should be added to the end of the purpose statement.
7. The introduction part is written in a way that is far from the systematic and language of the article. There is no specific flow. There is no standard in the tense used. It has been repeated in some places. The sentence in the previous paragraph is rephrased. Some sentences are inverted, incomplete and disconnected from the sentence before and after. In the introduction, general information is given first, but this information should not form the bulk of the introduction. Afterwards, the results of the studies done in this field are given in the literature and the meaning of this result and the studies done is explained and your reason for doing this study is linked to this section. Afterwards, the purpose and hypothesis are given and the introduction part is explained. Edit the entire entry from the beginning with this perspective.
8. Page4, line 113-144; You do not need to mention the work you have done in the past in the Participants section. The place of the data you have explained in Table 1 is in the results section. Age and gender data of individuals can be given, but it is not appropriate to give such large-scale data in the participants section. The presence of unnecessary information has made the manuscript text very complex. Edit this section. Add the flowchart of the participants.
9. Page 3, line 139; Express Chi² as x² in Table 1 and write its explanation under the table. Add the significance value of p under the table. Delete the dot-shaped items in the sub-headings. Separate the table into categorical and numerical variables and write the statistical analyzes under the table.
10. Page 4, line 146; You should have provided detailed information about the FCAI you used in the material section. You just wrote the question titles without giving any information. All this information should be added to the validity and reliability of this inventory, in what year it was created by whom, what the decrease in the score means and what the coat of arms means, how to answer it, whether there is a cut-off score or not. Some of the questions are open-ended, some are multiple-choice, and their details should be added and explained. How will the total score be made? How will they be analyzed? Please explain very comprehensively.
11. The material and method section was not written in a specific systematic. Add the title of the study design to the beginning of this section. Under this heading, you should add the information on how you determined the number of individuals participating in the study. Write the information about how and what you do the power analysis with reference, together with the source. You should also give information about the general design of the study in this section. The material and methods section is very vague and little is known about study design. You should explain this part in great detail.
12. Information on where and when the ethics committee was obtained should be added to the study design section. In addition, the information on the date on which the study data was collected should be added. In addition, information on who made the assessment and how many years of experience the person making the assessment has in this field should be added to this section.
13. You should add information about whether the data is normally distributed and whether these linked data are parametric or non-parametric in the data analysis section. Also, you did not specify how you analyzed the categorical variables. Add this information.
14. In the results section, add the p values where you do not find a significant difference.
15. The discussion was not made in accordance with the article writing system. You used too many personal comments. You have given little space to literature knowledge and results. What you had to do in this section was to relate your results with literature sources and make a comment from there. The available resources are very scarce.
16. What were the strengths and weaknesses of the research? What were their limitations? Add these sections to the discussion section.
Reviewer 3 Report
Thank you for the opportunity to review this paper. The study involved the examination of the financial knowledge and judgment skills of adults with ADHD compared to those without ADHD and to explore the impact of income. The sample included 45 adults with ADHD and 47 adults without ADHD who were assessed using the Financial Competence Assessment Inventory. Results showed that adults with ADHD had lower scores in areas such as awareness of bill arrival, knowledge of income, creating a reserve fund, and understanding of assets, compared to those without ADHD. However, no relationship was found between income and financial knowledge. Overall, the authors did a nice job of presenting the issue, describing why it is important, and detailing their analysis. The discussion also does a good job of summarizing the findings, identifying limitations, and calling attention to the need for supporting adults with ADHD in financial functioning.
I have the following feedback/questions for the authors:
Why was the FCAI selected as the measure for use in this study? Is the FCAI a valid assessment tool? Please provide some information on its psychometric properties.
Please also add some detail regarding the validity of the other measures used, including the WURS-K and ADHD-SR.
Please check formatting. Left align and remove the bullet points for line items in the first column of Table 1.
I appreciate the list of and details on the 39 items of the FCAI, but the presentation of the items seems choppy. This content may be better displayed in a table.
Left align the items in the first column of Table 2.
There is missing data from one participant with ADHD. Please describe how missing data was handled for your analysis.
Left align the items in the first column of Table 3.
There appears to be some minor grammatical and language errors throughout. Additional copyediting is recommended to further improve the overall presentation of the paper.
I wish the authors all the best in their continued work in this area.
Round 2
Reviewer 2 Report
Reviewer Comment
While some of my earlier edits seem to have been made, there are still many places in the manuscript that you need to correct. The title should be changed and a title that better reflects the content of the work should be created. There are a few details in the summary and introduction that you need to fix. The method section should be organized as general titles and content, and the study design section should be explained in detail. The scales should be explained in much more detail, and the data analysis section should be arranged in terms of written language. In the conclusion section, there are missing information in the tables and the text, which should be completed. The weakest part of the manuscript is the discussion section. An article is available without much literature support. too many personal comments. Literature resources are still scarce. The entire discussion section should be revised. In addition, the sources you use in the manuscript are old, much more current sources should be used. Please make all these suggestions carefully and completely.
Revision
1. The title should be changed. The title you use is very general and does not show what you are examining in the study. “DEHB'li yetiÅŸkinlerin günlük finansal bilgi ve muhakeme becerilerinin incelenmesi” ya da “DEHB'li yetiÅŸkinlerin günlük finansal bilgi ve muhakeme becerilerine iliÅŸkin güçlü ve zayıf yönlerinin incelenmesi” either of these can be revised for the title.
2. Include the age range, mean and standard deviation of the individuals participating in the study in the method section of the summary.
3. Page 2, line 40; “[e.g., 2–6]” Delete e.g., just write reference numbers.
4. Page 2, line 71-79; “Importantly, the studies listed here, which were all performed in selected samples, are corroborated by a large-scale Swedish population study using objective financial data [23]. This study demonstrated that adults who were ever diagnosed with ADHD were more likely to incur arrears which appeared to be related to receiving new consumer credit less often, despite asking for more credit, than the general population. Particularly, adults with ADHD were four times more likely to accumulate arrears regarding unpaid educational support, road taxes and alimony, misuse of bank accounts (e.g., overdrafts), and impounding of property as compared to the general population [23].” Since you use the same source in consecutive paragraphs, the reference number you use at the end of the first paragraph is not necessary. It is sufficient to use the source number 23 once, at the end of the last sentence.
5. Page 2, line 80-82;“To deal with financial difficulties, general education programs aimed at improving financial knowledge and skills are often offered and are considered to be an antidote to dealing with financial issues and financial complexity by many policymakers.” If you are using a phrase that is often presented, you should include references to these sources, which are more than one, at the end of the sentence.
6. Page 2, line 82-84; “A recent systematic review and meta-analysis, however, demonstrated that a one-size-fits-all financial education does not result in improved financial behavior [24].” The location of the 24th source is incorrect. After this paragraph, you continued to describe the meta-analysis study, but did not use any references at the end of the sentence. Here you should put the source in the last place where the information about the meta-analysis study ends.
7. Page 3, line 127; The name of the design section should be changed to study design. Your explanations in this section are very inadequate. In this section, information such as the general plan of the study, its duration, the place and date of the ethics committee approval, and the place where the study was conducted should be given. Rearrange this section.
8. Add the age range information of the individuals to the participants section of the method.
9. Page 3, line 129; Was power analysis performed to determine the number of individuals in the participants section? If so, how was it done? This information should be added. In addition, a flowchart figure containing the number of participants should be added to this section.
10. Page 4, line 152; In the material section, add the FCAI's internal reliability coefficient. Explain who developed the questionnaire, how the scoring is done, what it means to increase and decrease the score, and if there is a cut-off point.
11. Page 8, line 164; WURS-K and ADHD-SR. Open separate titles about the scales and explain the scales in more detail. Explain by whom it was developed, how is the scoring, what does the increase in the score mean, what are the cut-off scores, what are the validity and reliability coefficients.
12. In the material section, open the sociodemographic information form title and indicate what information about the individuals was collected.
13. Page 8, line 171; The procedure title is unnecessary, provide information about the Plan in the study design section.
14. Page 8, line 177-181; “The Statistical Package for the Social Sciences version 28.0 will be used for analyses. The normality of continuous variables (i.e., age, years of education, WURS-K, and ADHD-SR) will be checked. When these variables have a normal distribution, parametric t-tests will be used for group comparison. In case of non-normal distributions, non-parametric Mann-Whitney U tests will be applied. Chi-square tests will be used to compare adults with and without ADHD on categorical variables.” There is an incorrect pronunciation in the time statement in your spelling language. You should use the language when it was done. At this stage, why did you use such an expression when it was clear whether the data were normally distributed or not? Edit this section carefully.
15. Page 9, line 198-202; “The clinical and descriptive characteristics of both groups are presented in Table 2. Both groups had a similar age and completed a similar number of years of education. Furthermore, no differences were found between groups regarding the number of males and females. As expected, adults with ADHD showed significantly higher scores on the WURS-K and ADHD-SR than adults without ADHD (Table 2).” Add the p values.
16. Page 9, line 218; “For all other items of the FCAI, no significant differences were found between adults with and without ADHD.” Add the p values.
17. Tablo 3’ün altına hangi istatistiksel analiz yöntemini kullandığınız bilgisini ekleyiniz.
18. Page 12, line 236; Within the group of adults with ADHD, no significant differences were found between the relatively low and relatively high-income groups (Table 4). Add the p values.
19. Under Table 4, add the significance value of p and the statistical method used. In addition, put a sign next to the significant values of p in the table to make it more understandable.
20. Page 13, line 266; In this section, you gave your results and then made a comment by expressing your personal views. Debate cannot be made this way. You should relate your results to the literature and make your comments based on literature knowledge, away from your personal interpretations. This shortcoming is evident throughout the discussion section. You should write the entire discussion this way using much more literature sources.
21. There are a lot of old sources that you use in general. In particular, you should write the introduction and discussion section with the most up-to-date sources. Update your sources.
